# Human Gut Microbiome Response Induced by Fermented Dairy Product Intake in Healthy Volunteers

**DOI:** 10.3390/nu11030547

**Published:** 2019-03-04

**Authors:** Olesya Volokh, Natalia Klimenko, Yulia Berezhnaya, Alexander Tyakht, Polina Nesterova, Anna Popenko, Dmitry Alexeev

**Affiliations:** 1PepsiCo R&D, Inc., Leningradsky prospekt, 72-4, 125315 Moscow, Russia; olesyavolokh@gmail.com (O.V.); yulia.berezhnaya@pepsico.com (Y.B.); polina.nester@gmail.com (P.N.); 2Knomics LLC, Skolkovo Innovation Center, Bolshoy bulvar street 42-1, 143026 Moscow, Russia; natasha.klmnk@gmail.com (N.K.); popenko@atlasbiomed.com (A.P.); dmitry.g.alexeev@gmail.com (D.A.); 3Institute of Gene Biology of Russian Academy of Science, Group of Molecular Organization of Genome, 34/5 Vavilova Str., 119334 Moscow, Russia; 4ITMO University, Computer Technology Department, Kronverkskiy pr., 49, 197101 St. Petersburg, Russia; 5Atlas Biomed Group, 92 Albert Embankment, Lambeth, London SE1 7TT, UK

**Keywords:** gut microbiota, probiotics, clinical trial, fermented dairy products, responders

## Abstract

Accumulated data suggests that the gut microbiome can rapidly respond to changes in diet. Consumption of fermented dairy products (FDP) fortified with probiotic microbes may be associated with positive impact on human health. However, the extent and details of the possible impact of FDP consumption on gut community structure tends to vary across individuals. We used microbiome analysis to characterize changes in gut microbiota composition after 30 days of oral intake of a yoghurt fortified with *Bifidobacterium animalis* subsp. *lactis* BB-12. 16S rRNA gene sequencing was used to assess the gut microbial composition before and after FDP consumption in healthy adults (*n* = 150). Paired comparison of gut microbial content demonstrated an increase in presence of potentially beneficial bacteria, particularly, *Bifidobacterium* genus, as well as *Adlercreutzia equolifaciens* and *Slackia isoflavoniconvertens.* At a functional level, an increased capacity to metabolize lactose and synthesize amino acids was observed accompanied by a lowered potential for synthesis of lipopolysaccharides. Cluster analysis revealed that study volunteers segregated into two groups with post-intervention microbiota response that was dependent on the baseline microbial community structure.

## 1. Introduction

The majority of human gut microbes belong to Firmicutes and Bacteroidetes phyla, with Actinobacteria phylum being a minor but essential component of the gut microbiome [1]. *Bifidobacteria* belonging to this phylum have been known to be essential and beneficial inhabitants of the human gut long before the era of molecular-genetic technologies. The clade plays important roles in vitamin production, protection against pathogens, regulation of immune system and lactose utilization. *Bifidobacteria* confer functional benefits by cross-feeding other members of gut microbiota specializing on production of butyrate, an essential substance for colon epithelial cells with anti-inflammatory and anti-cancer properties [2]. Low levels of bifidobacteria are associated with various adverse clinical conditions [3,4,5]. Its abundance can be increased by consuming fermented dairy products (FDP) containing live microbes, probiotic supplements or by supporting bifidobacteria with prebiotics [6]. Introduction of probiotic strains of bifidobacteria to human gut has been reported to improve clinical status in diseases like antibiotic-associated diarrhea [7,8], necrotizing enterocolitis [9], chronic pouchitis [10]. Improvement has also been reported in allergic diseases including atopic eczema [11,12] allergic rhinitis [13] and allergic diarrhea [14].

With the advent of 16S rRNA gene sequencing as a routine scientific method, it is now possible to investigate microbiota-mediated impact of probiotic strains of *Bifidobacteria* and probiotic-fortified food products on human gut microbiome in more detail. Particularly, the approach allows efficient characterization of interactions between probiotic microbes and gut microbial species. Although recent culturomics efforts have succeeded in capturing the majority of gut microbial species [15], in clinical studies the 16S rRNA gene sequencing is still superior to cultivation-based approaches as it allows to obtain information about the total community composition in an economic and high-throughput way. A recent survey demonstrated that intake of probiotic-fortified fermented milk products not only decreased levels of commonly associated gut pathobiont species but also directly improved the production of short chain fatty acids (SCFA)—possible biomarkers associated with healthy gut function [16]. Interestingly microbiome changes in studies of patients with irritable bowel syndrome paralleled improvement in reported disease symptoms [16,17]. Another probiotic dietary intervention study revealed that the persistence of probiotic strains in the gut of healthy subjects depends on the initial composition of the microbiota [18]. Prediction of the individual response of gut microbiota to specific probiotics may allow development of personalized nutrition schemes intended to promote or maintain human wellbeing.

Here we examined the effect of a fortified fermented dairy product (FDP) intake on human gut microbiota composition using 16S rRNA sequencing of stool samples. In this controlled study, volunteers consumed FDP for 30 days; clinical data and stool samples were collected on the first and last days of the study.

## 2. Materials and Methods

### 2.1. Study Design

The study was a part of a large open prospective controlled study evaluating the efficacy and tolerability of fermented dairy products as well as the effect of product consumption on gut microbiota in healthy volunteers. The research was approved by a local ethics committee of Alliance Biomedical—Russian Group, Ltd. The experimental group included 150 subjects. The inclusion and exclusion criteria are listed in the Appendix A section. The sample size was initially determined for the outcomes which are out of the scope of this study—the number of volunteers with improved GI health status according to clinical laboratory indicators. The minimum effect size of microbiota composition change that can be detected with given number of samples (*n* = 150), statistical power of 80% and significance level 0.05 was calculated using a framework for PERMANOVA power estimation [19] and was equal to ω^2^ = 0.007. 

All participants signed informed consent before the start of the study. Status of the volunteers was assessed during three site visits: Visit 0 (screening)—the day of study enrollment; Visit 1—study day 2 and FDP consumption start day; Visit 2 was on the day 15 of the study (14 days from start of FDP consumption); Visit 3 was on the day 31 (30 days after consumption start day) and it was the last consumption day. In order to assess test article consumption compliance and reveal any adverse events, two telephone calls were performed: on days 8 and 21 of the study. Clinical status assessment, anthropometric measurements, thermometry and physiological assessments were conducted (see Appendix A). Stool samples were collected before the dietary intervention (Visit 1) and immediately afterwards (Visit 3). Possible adverse/serious adverse events, compliance and adherence to the dietary recommendations were assessed during Visits 2,3 and telephone calls. All subjects’ visits were performed on an outpatient basis. 

For 30 days, the volunteers consumed fermented milk product—a yoghurt fortified with *Bifidobacterium animalis* subsp. *lactis* BB-12—125 mL in the morning and 125 mL in the evening daily. They also followed dietary recommendations in an uncontrolled setting—diet #15 according to Nomenclature of Pevzner Diets (detailed menu suggested as a proper dietary plan for healthy subjects [20]). The volunteers did not take any prescription medications (except hormonal contraceptives for women) or biologically active supplements.

### 2.2. Sample Preparation and Microbiome Data Analysis

Amplicon sequencing of V4 variable region of microbial 16S rRNA gene was performed on an MiSeq sequencer (Illumina, San Diego, CA, USA) (see Appendix A). Raw sequencing data is deposited in European Nucleotide Archive under project accession number PRJEB26974. Data were analyzed in QIIME 1.7.0 [21]. Taxonomic analysis was performed by reference-based classification using uclust_ref algorithm and the HITdb 1.0 database [22] at the level of operational taxonomic units (OTU) with 97% sequence identity threshold. Prior to the analysis, HITdb 1.0 database was preprocessed using TaxMan software [23] to obtain non-redundant database of sequences that can be identified by our primers. In the resulting database, the OTUs with ambiguous classification at some taxonomic rank were marked using “/” (for example, belonging to “*Blautia*/*Roseburia*” genus). The classified reads for each sample were randomly rarefied to the same number (9000 reads per sample); samples with lower coverage were not included in the analysis. Additional validation was performed using Greengenes 13.5 database [24] using the same sequence identity threshold. Estimation of alpha-diversity for each sample was performed using QIIME and three metrics: “chao1”, “PD_whole_tree”, and “shannon”. Beta-diversity (pairwise dissimilarity between the gut community structures) was estimated using Bray-Curtis metric. Read counts of microbial species, genera and families were calculated as the sum of reads assigned to the OTUs belonging to the respective taxon. At each taxonomic rank level, the total relative abundance was normalized to 100% for each sample. Prediction of metabolic potential profiles was performed using PICRUSt [25]. Statistical analysis of the taxonomic composition vectors was performed in R statistical programming language, version 3.3.0 (see Appendix A). Exploratory data analysis was performed using Knomics-Biota online platform [26] (interactive analytical report is available online at [27] (basic report), [28] (paired report), project ID 302).

### 2.3. Analysis of Responders

Coefficient of change in the relative abundance for each lactose-fermenting microbial taxon (LFT) was calculated for each subject *i*:(1)$$d_{ij}=\frac{2\cdot (a_{ij}-b_{ij})}{\left(a_{ij}+b_{ij}\right)},$$
where *a_ij_* is the abundance of LFT *j* in the sample of subject *i* before FDP consumption; *b_ij_*—the abundance of LFT *j* in the sample of subject *i* after FDP consumption.

If both abundance values of a taxon before and after the FDP consumption were equal to zero, then *d_i_*_j_ was set to 0. Pairwise dissimilarity between the subjects according to the extent of changes in LFT analysis was calculated by applying Euclidean metric to the vectors of changes for all LFT. Cluster analysis of samples was performed using k-means algorithm. The optimal number of clusters was selected as the one that provides highest average silhouette width (ASW = 0.20 for two clusters).

Comparison of initial microbiota composition between the responders and other subjects was performed using MaAsLin algorithm [29] with the following parameters: enabled arcsin square root transformation and identification of outliers, without boosting. Multiple testing adjustment was performed using the Benjamini-Hochberg method and associations were considered significant if the adjusted p-value was less than 0.05.

The associations between physiological factors and responder/non-responder status were accessed using Mann-Whitney test for numeric factors (age, BMI, arterial pressure, cardiac rate, body temperature) and chi-squared test for categorical factors (gender).

## 3. Results

16S rRNA gene sequencing yielded 34,663 ± 9641 reads per sample. Sequencing statistics and metadata are listed in the Appendix A. The fraction of the identified reads was 96.6 ± 2.6% confirming the high quality of the sequencing data and applicability of the selected classification algorithm (see Methods). In total 54 families, 126 genera and 519 species were detected in at least one sample (see Appendix A). Analysis of microbial community richness dynamics for each volunteer showed that there was no significant change in alpha-diversity after FDP intake (Shannon index 5.6 ± 0.6 and 5.6 ± 0.1, *p* = 0.68, paired Welch’s test).

Individual microbial species cooperate with the other species within the gut community, making trophic chains and other connections, thus forming symbiotic subcommunities (cooperatives) [30]. In order to determine the effect of FDP intake at this level, the cooperatives were identified based on correlation analysis of the abundance of microbial genera for all samples corresponding to Visits 0 and 3 (see Methods). As a result, five large potential cooperatives were identified along with a few smaller ones (see Figure 1). Complete sets of genera in each cooperative are listed in Appendix A.

Comparison of overall microbial community structure before and after FDP consumption adjusted for the significantly associated factors showed that the gut microbiota composition of volunteers was significantly changed (PERMANOVA test, Bray-Curtis metric on the level of genera, *p* = 0.006, R^2^ = 0.51%), although the degree of the change was moderate—on average 1.4 times lower than the mean group variability level (Bray-Curtis measure: 0.41 ± 0.14 between the paired samples vs. 0.53 ± 0.11 between all possible samples, *p* = 0, Welch’s test).

Paired comparison of the abundance of individual microbial taxa before and after the intervention using metagenomeSeq [31] (see Appendix A) revealed 39 taxa as significantly increased and 24, decreased (see Appendix A). At a family level, Coriobacteriaceae, Bifidobacteriaceae, Staphylococcaceae, and Erysipelotrichaceae, increased their fraction. Moreover, the list of significantly increased genera and species included *Bifidobacterium* (*B. bifidum*, *B. adolescentis*, *B. animalis*, *B. bifidum*, *B. longum*)*, Adlercreutzia* (*A. equolifaciens*), *Slackia* (*S. isoflavoniconvertens*), *Collinsella* (*C. aerofaciens*), *Catenibacterium* (*C. mitsuokai*), *Streptococcus* (*S. thermophilus/vesti**bularis*), and other taxa. However, only two genera and one family decrease (*Lachnoclostridium/unclassified* and *Roseburia* genera and Acidaminococcaceae family) were associated with the FDP consumption. Interestingly, when a similar analysis was performed at the level of microbial cooperatives, no significant changes were detected for any of the cooperatives. 

The impact of the FDP intake on microbiota of the volunteers was also assessed at the level of functions—via the analysis of changes in relative abundance of metabolic pathways (see Methods). In total two pathways were significantly increased and 24, decreased (see Appendix A). The pathways with the most profound changes (for which the highest fraction of genes were affected) included an increased “Phosphotransferase system (PTS)” pathway—the transport systems specific for the Firmicutes phylum—and also the decreased pathways “Bacterial chemotaxis” and “Flagellar assembly”—reflecting the effects of the decreased Gram-negative:Gram-positive microbes ratio after the FDP consumption. Moreover, there was an increase in pathways associated with starch and simple sugars transport and amino acids synthesis. At the module level (see Appendix A), FDP intake was associated with increased lactose transport system genes (PTS system, lactose-specific II component)—in agreement with the observed fraction of lactose-fermenting bacteria at the taxonomic level. Among the decreased modules, there is a module related to the synthesis of lipopolysaccharides (Lipopolysaccharide biosynthesis, KDO2-lipid A)—immunogenic components of Gram-negative bacteria cell walls.

Response of gut microbiota community structure to intake of probiotics can vary across individuals [18]. In order to explore variation of response to FDP intake on subject level for our cohort, firstly we assessed the changes in relative abundance of the major target group of microbes that was expected to react—lactose-fermenting taxa (LFT): the list included *Bifidobacterium, Lactobacillus*, *Lactococcus*, *Streptococcus*, *Slackia*, *Corynebacterium,* and unclassified *Enterobacteriaceae* (see Methods). Cluster analysis of the subjects’ microbiomes before the FDP course based on the extent of changes in total LFT abundance showed that the group of volunteers formed two clusters (ASW = 0.2)—cluster #1 (*n* = 75 subjects) and cluster #2 (*n* = 58). Therefore, subjects can be divided into two groups in which the pool of LFT demonstrated two different types of response to FDP consumption. For subjects from the cluster #1, the microbiota manifested a significantly weaker increase of the levels of LFT in comparison with the cluster #2 (change −0.10 ± 1.20% vs. 0.51 ± 1.26, respectively, *p* = 0, Welch’s test).

Next, we compared the clusters by the change in total taxonomic composition (not just LFT) after FDP consumption. The total taxonomic composition for members of cluster #1 did not change significantly (PERMANOVA test, *p* = 0.0813, R^2^ = 0.61%), while for cluster #2 the change was significant (*p* = 0.0004, R^2^ = 2.07%). Further, changes in composition were compared between clusters on a more detailed level, for individual taxa and cooperatives; results suggest that the two clusters are different by the type of response of not just LFT but also the other microbial taxa (see Figure 2, Appendix A).

Cluster #1 showed fewer changes (*n* = 9 taxa/cooperatives), the most pronounced of which was a decrease of *Lactococcus* genus abundance (including *L. plantarum/raffinolactis*). For cluster #2, the number of affected taxa/cooperatives was higher (*n* = 108). Therefore, cluster #2 can be considered responders in comparison with the other samples (i.e., cluster #1) (Figure 2).

After the responders were determined as members of cluster #2, we sought to identify distinctive features of the responders’ microbiota that might be predictive in the general population. For this purpose, baseline microbiota composition (before intervention) was compared between responders and non-responders using the MaAsLin method (see Methods), results are listed in Table 1.

These results show that the microbiota of responders contains a lower fraction of lactose-fermenting taxa, while the fraction of taxa from cooperative #2, particularly, the members of Bacteroidaceae, is increased (Noteworthy, this effect cannot be explained solely by the compositionality of the microbiome data because the microbiota of the volunteers contains many other species besides LFT and members of cooperative #2). However, none of the physiological factors significantly differed between the two clusters at baseline.

## 4. Discussion

Semiquantitative microbiota composition profiles obtained using metagenomic analysis of stool samples give the most complete picture of gut microbial community structure independently of whether the species are cultivable. These profiles were used to estimate the change in gut microbiota composition after FDP consumption.

The lists of taxa differentially abundant before and after FDP consumption show significant overlap between multiple methods of statistical analysis, thus confirming the validity of the findings. Among the decreased taxa, there are various species from the Firmicutes and Bacteroidetes phyla, usually comprising up to 90% of total bacterial abundance in gut microbiota of healthy people [32]. At the same time, there was a pronounced increase in the abundance of the third most dominant phylum, Actinobacteria, including *Bifidobacterium*. Many members of this genus are probiotic microorganisms, and their role in anti-inflammatory activity, protection from pathogenic microorganisms and vitamin production has been noted by others [33]. Interestingly, recent study has shown that this genus commonly associated with infant microbiota can also dominate microbiome of adult individuals from some world populations [34,35]. The *Bifidobacterium* is among the genera that increased their abundance significantly after FDP consumption. Noteworthy, in addition to the increase of *B. animalis* fraction, there was also a significant increase in the abundance of another bifidobacteria—*B. bifidum, B. adolescentis, B. animalis, B. longum*—that were not included in the FDP starter culture composition. This indicates that the FDP consumption not just leads to increased presence of *B. animalis* due to its direct introduction but also potentially affects the ecology of the gut microbiome by supporting its resident bifidobacterial species. The *Streptococcus* genus (including *Streptococcus thermophilus*, a component of the starter culture) was also increased in abundance. Further studies including a control group consuming a placebo product with identical formulation but lacking the probiotic are required to dissect the effects of the probiotic from the effects of the fermented product itself. Additionally, microbiome analysis of stool samples collected several weeks after the end of FDP consumption will provide clues to assess the persistence of the observed shifts in species populations.

Interestingly, there was an increase of other *Actinobacteria*, including several species of the *Coriobacteriaceae* family. This effect could be attributed to the increased levels of lactose in the diet provided by regular intake of the test product. These taxa have a specific ability to metabolize lactose to lactate; meanwhile, a part of the lactose originally derived from dairy products remains intact during preparation of FDP, is incompletely digested in small intestine and reaches the large intestine. *Actinobacteria* possess specific metabolic function that contribute to general human health by participation in metabolism of food components that increase antioxidant capacity. Specifically, *Adlercreutzia equolifaciens* and *Slackia isoflavoniconvertens* are active participants in isoflavone metabolism. The main dietary source of these substances are legumes, mainly soybeans that contain the isoflavones genistein and daidzein. These substances themselves are phytoestrogens, and a number of studies indicated an association between their consumption and improved reproductive functions, as well as with a reduced risk of breast cancer in women, and antioxidant properties [36,37,38]. *Adlercreutzia* and *Slackia* are capable of metabolizing daidzein into equol [39]. Equol is an isoflavandiol manifesting phytoestrogenic activity with a potentially positive effect on human health, including hormonal and cardiovascular functions [40] and anticancer activity [41].

The ability to metabolize isoflavones into equol is a quite specific microbial feature: it is estimated that only about a third of the world population harbor such microbial species in their gut. Thus, a hypothetical therapeutic diet with a high content of soybean products might not be effective for a large part of the population. Based on these facts, we can conclude that the observed increase of *Adlercreutzia* and *Slackia* genera after FDP consumption may improve the capacity of human microbiota for responding to a diet rich in isoflavones, including soy-based products. These observations suggest an opportunity to design food products and/or diets containing not only dairy components enriched with bifidobacteria and lactobacilli, but also soy.

The *Erysipelotrichaceae* family of bacteria were also significantly increased after FDP consumption (including species related to *Eubacterium dolichum* and *Catenibacterium mitsuokai*). While data on clinical associations for this bacterial family are ambiguous, there are a number of studies linking their increased abundance to inflammatory bowel diseases, as well as obesity, while other studies suggest opposite associations [42,43].

The groups of bacterial taxa detected during correlation analysis represent potential symbiotic cooperatives of species (Figure 1). Observed cooperatives vary by phylogenetic composition, and many features of their content are consistent with published data [44,45]. 

Half of cooperative #1 is formed by Clostridiales bacteria (*Eubacterium, Anaerostipes, Blautia, Dorea*) known as prominent producers of butyrate determining their anti-inflammatory activity and association with the healthy gut [46]. Studies of the effect of diet on microbiota composition show that levels of these taxa are increased when the diet is rich in fiber, commonly found in vegetables, cereals and other products [47]. 

The dominance of cooperative #2 is formed by members of *Bacteroidaceae* family including; *Bacteroides*, *Parabacteroides*, *Alistipes,* and others. Increased prevalence of these groups have been associated with the “Western diet”, a diet rich in animal fats, meat and sugars, as well as deficient in non-digestible dietary fiber [48]. Additionally, two members of a related family *Porhyromonadaceae* (*Odoribacter* and *Butyricimonas*) are present.

Cooperative #3 includes butyrate-producing clostridia associated with decreased risk of inflammatory bowel diseases as well as a number of distantly related microbes with symbiotic links. Among them are the microbes suggested to be significant in the regulation of weight and dietary behavior. The *Christensenella* genus (a representative species is *C. minuta*) is the most inheritable gut microbe; it is also associated with normal weight and prevents obesity in mice models [44]. The *M. smithii* archaeon facilitates a more efficient fermentation of dietary fiber and may contribute the regulation of weight and dietary behavior. *Methanobrevibacter* and *Christensenella* were previously observed to be inherited together; both are associated with normal BMI [44,49], although the mechanisms underlying this co-occurrence have not been identified yet.

Generally, none of the microbial cooperatives significantly changed their abundance as the result of FDP consumption. This difference from the results obtained during species-level analysis may be due to the observation that many of the differentially abundant species are not included in any of the cooperatives (only large cooperatives were examined). Secondly, it may be related to the fact that FDP consumption represents a relatively small change in total daily dietary intake of volunteers—in comparison with changing one’s diet to follow certain recommendations, e.g., aiming to increase total dietary fiber consumption by including more fruit, vegetables, and whole grains into diet [50].

However, there were significant changes for the cooperatives when subjects were stratified into responders and non-responders (Figure 2). Namely, responders had increased baseline levels of cooperative #2 enriched in *Bacteroidaceae* and decreased—of cooperative #1 enriched with Clostridiales. The taxa included in cooperative #2 are reported to be associated with long-term “Western diet” [48]. While no significant associations between responder/non-responder clusters and physiological factors were identified, one can speculate that FDP consumption may have a more profound positive impact on the microbiota of individuals on “Western diet”.

In addition to the evaluation of impact of FDP consumption on species-level composition, microbiota analysis allowed evaluation of impact on microbiota functions by investigating selected changes in metabolic potential. Significant functional changes were observed reflecting increased capacity of the community to metabolize lactose, other simple sugars, starch, as well as to synthesize amino acids. This was accompanied by a decrease in synthesis of immunogenic molecules (lipopolysaccharides) that can be interpreted as a reduction of the proinflammatory potential of the microbiota.

## 5. Conclusions

Significant shifts in gut microbial taxonomy and function suggest that a single month of FDP consumption may promote general positive effect on human gut and possibly total host physiology. Further study is required to confirm any potential lasting impact on indigenous microbiota and other resident microbes after FDP discontinuation. Together with the interesting effect of increasing equol-producing bacteria, the results suggest potential for multi-faceted positive impact of FDP consumption on human gut microbiota by promoting shifts in microbiota species that are associated with positive impact on biomarkers commonly associated with inflammatory, hormonal, and cardiovascular function.

The identified gut microbial signature of responders requires confirmation as well as further investigation regarding generalizability to the general population. Microbiota analysis may help identify optimal probiotics to support personalized nutrition recommendations based on individual gut microbial community structure and function.

## Figures and Tables

**Figure 1 nutrients-11-00547-f001:**
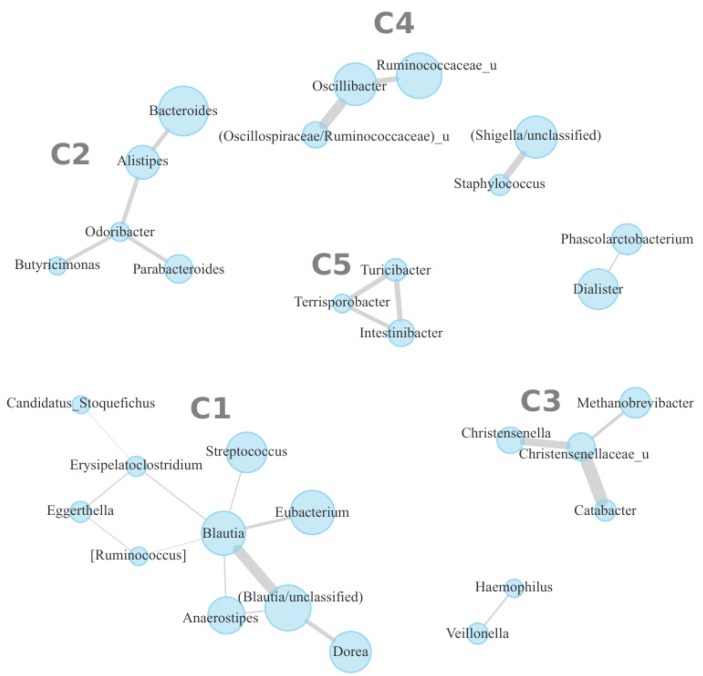
Co-occurrence graph of microbial genera in the samples of the volunteers. Vertices denote the genera; the size of each vertex is proportional to the average abundance of the genus across all samples. The thickness of edges is proportional to the absolute value of correlation coefficient. The “_u” postfix denotes all unclassified genera from the respective family. C1–C5 denotes cooperatives # 1–5, respectively.

**Figure 2 nutrients-11-00547-f002:**
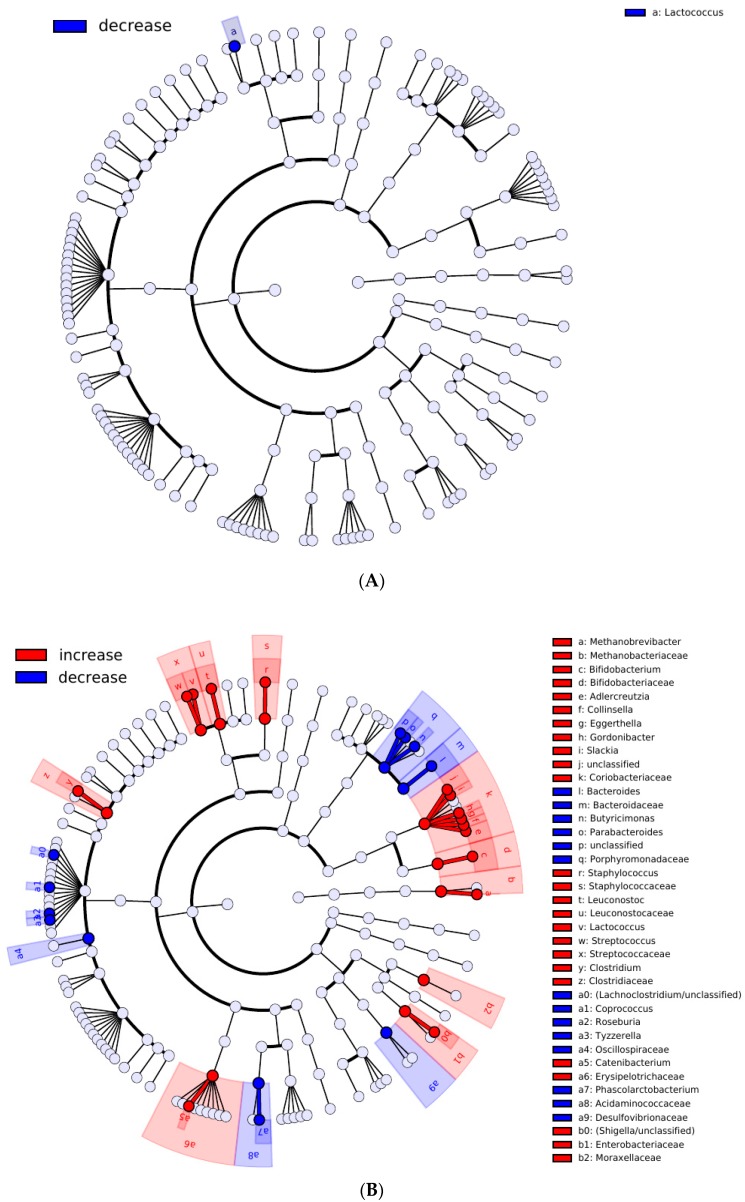
Microbial taxa down to genera level, relative abundance of which significantly changed after the course of FDP consumption, stratified by two clusters. (**A**) Cluster #1 (non-responders); and (**B**) cluster #2 (responders). The hierarchical visualization is performed using GraPhLan: The increased taxa are shown in blue, the decreased in red.

**Table 1 nutrients-11-00547-t001:** Taxa and cooperatives differentially abundant in the gut microbiota of responders and non-responders before the FDP consumption. For each feature, beta-coefficient of linear model is shown. (**A**) Increased in responders; and (**B**) decreased in responders.

(**A**) Increased in responders.				
**Taxon**	**Rank**	***p***	**Adjusted *p***	**Linear Regression Coefficient**
Cooperative #2	Cooperative	0.0038	0.0097	0.0692
*Bacteroidaceae*	Family	0.003	0.0393	0.068
*Oxalobacteraceae*	Family	0.0033	0.0393	0.0031
*Bacteroidales*	Order	0	0.0007	0.1074
*Bacteroidia*	Class	0	0.0005	0.1074
(**B**) decreased in responders				
**Taxon**	**Rank**	***p***	**Adjusted *p***	**Linear Regression Coefficient**
Cooperative #1	Cooperative	0.0003	0.0013	−0.0954
*Streptococcaceae*	Family	0.0001	0.0417	−0.0433
*Coriobacteriaceae*	Family	0.0046	0.0375	−0.0192
*Peptostreptococcaceae*	Family	0.0024	0.0392	−0.0204
(Lachnospiraceae/unclassified)	Family	0.0052	0.0417	−0.0215
*Lactococcus*	Genus	0	0	−0.0222
*Blautia*	Genus	0	0.0013	−0.0543
*Lactococcus_lactis*	Species	0	0.0004	−0.0185
OTU1383 (*Blautia*)/OTU1528 (*Blautia*)/OTU819 (*Blautia*)	Species	0	0.0014	−0.0425
OTU262 (*Ruminococcus*)/OTU286 (*Ruminococcus*)	Species	0	0.0098	−0.0530
OTU513 (*Blautia*)/OTU661 (*Blautia*)	Species	0.0003	0.0324	−0.0244

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
