# Peer review of "Human Gut Microbiome Response Induced by Fermented Dairy Product Intake in Healthy Volunteers"

_nutrients, 2019, doi:10.3390/nu11030547_

Round 1

Reviewer 1 Report

Line49: It must be noted that recent advances have indicated that more than 90% of the gut microbiota is cultivable. I believe the approach of Culturomics needs to be acknowledged before making that statement.

In the results section. Please mention Figure number corresponding to the appropriate result. 

Author Response

1. We appreciate your comments on our manuscript. The language was additionally edited.

2. Thank you for this valuable reсommendation. We added a sentence with a reference to a recent large-scale advances in cultivating gut microbes (lines 52-56).

3. We added a reference to the Figure 1 in lines 156 and 296. The Figure 2 was referenced to in line 213. We also added the reference to it in the lines 222 and 326.

Reviewer 2 Report

This manuscript describls a clinical trial where a fermented product containing B. animalis subsp lactis was administered to 150 subjects. The subjects followed a particular dietary recommendation for the duration of the trial. Pre and post trial fecal samples were collected for analysis. Microbiota compositional analysis identified that the subject participants formed 2 groups with respect to the microbiota composition following the trial period and the authors attempt to explain these compositional changes while being cognisant of the drabacks of this study (no placebo group, responses due to the fermented product or probiotic). This study is valuable in that B. animalis subsp lactis is incorporated in many fermented milk products as a probiotic. The manuscript is well written, the results are clearly presented and I believe this manuscript is of value to both the scientific and nutrition communities.

Author Response

We appreciate your comments on our manuscript. The language was additionally edited.

Reviewer 3 Report

This work demonstrates the functionality of fermented food products via metagenomic analysis. The data analysis method reflects the current trend of lactic acid bacteria research. Results and discussions are well organized. 

Author Response

Thank you for reading and giving positive comments on our manuscript.